# Full spectrum fluorescence lifetime imaging with 0.5 nm spectral and 50 ps temporal resolution

Gareth O. S. Williams [1], Elvira Williams[2], Neil Finlayson[3], Ahmet T. Erdogan [3], Qiang Wang [1], Susan Fernandes[1], Ahsan R. Akram [1], Kev Dhaliwal[1], Robert K. Henderson[3], John M. Girkin [2✉] & Mark Bradley [4✉]

The use of optical techniques to interrogate wide ranging samples from semiconductors to biological tissue for rapid analysis and diagnostics has gained wide adoption over the past decades. The desire to collect ever more spatially, spectrally and temporally detailed optical signatures for sample characterization has specifically driven a sharp rise in new optical microscopy technologies. Here we present a high-speed optical scanning microscope capable of capturing time resolved images across 512 spectral and 32 time channels in a single acquisition with the potential for ~0.2 frames per second (256 × 256 image pixels). Each pixel in the resulting images contains a detailed data cube for the study of diverse time resolved light driven phenomena. This is enabled by integration of system control electronics and on-chip processing which overcomes the challenges presented by high data volume and low imaging speed, often bottlenecks in previous systems.

[1] Centre for Inflammation Research, Queen's Medical Research Institute, University of Edinburgh, 47 Little France Crescent, Edinburgh EH16 4TJ, UK. [2] Centre for Advanced Instrumentation, Department of Physics, Durham University, South Road, Durham DH1 3LE, UK. [3] School of Engineering, Institute for Integrated Micro and Nano Systems, University of Edinburgh, King's Buildings, Alexander Crum Brown Road, Edinburgh EH9 3FF, UK. [4] School of Chemistry, University of Edinburgh, David Brewster Road, Edinburgh EH9 3FJ, UK. ✉email: j.m.girkin@durham.ac.uk; Mark.Bradley@ed.ac.uk

Unmixing of signals from complex fluorescent samples[1] is enhanced by high-resolution time-resolved emission spectroscopy (TRES[2]), where the optically efficient, simultaneous acquisition of full emission spectrum and lifetime datasets enables complete exploitation of the fluorescence signal and determination of small changes in emission profiles. The emission properties of individual fluorophores in complex environments such as biological samples are affected by a host of inter-molecular interactions and environmental fluctuations including resonant energy transfer, pH, viscosity, temperature, and a range of quenching pathways that lead to shifts in both emission spectrum and lifetime of excited states[3]. When multiple endogenous and exogenous emitting species are present there are further complex interactions leading to subtle shifts in emission, often only over a few nm.

Full-spectral fluorescence lifetime imaging microscopy requires a volume and speed of data collection which has constrained previous systems to a small number of parallel spectral channels[4–14]. This has inherently limited the complexity of the signals that can be distinguished by such systems, reducing their ability to monitor subtle spectral or lifetime changes. Wide ranging influences can introduced perturbations in spectral signals, such as pH or, in the case of tissue, structural or biomarker influences such as those found in cancer[15–18]. Some commercial lifetime imaging systems, such as the Leica Stellaris, can achieve increased spectral resolution through multiple rapid sequential image captures, with incremental shifts in detection wavelength. However, the requirement for multiple image acquisitions slows down the entire acquisition process and contributes to detrimental effects such as photobleaching and loss in image quality due to sample motion. An optimal solution is to obtain the entire wavelength-lifetime spectrum for each image pixel in a single pass. Whilst previous work has utilized line arrays of SPADs[4,19], including the ability to time gate arriving photons on the sensor[20], the implementation presented here offers key advantages in addition to the very high number of spectral channels. Firstly, the ability to generate lifetime histograms "on-chip" greatly reduces the volume of data typically required for TCSPC lifetime imaging. This enables greater photon number processing throughput, resulting in increased signal to noise ratios, allowing the use of lower excitation light intensities and reduced imaging

exposure times. To increase dynamic range of the sensor, time-bins can be chained together in pairs (resulting in 16 bins) increasing the maximum signal per sensor pixel from 1024 to over 2 million (10 bit vs 20 bit)[21]. Here, chaining was used to allow for the bright emission peaks to be captured along with the edges of the spectrum. Secondly, all of the photon timing electronics are contained on-chip removing the requirement for complex printed circuit board electronics and delay lines. The chip is packaged onto a single Printed Circuit Board (PCB) containing a Field Programable Gate Array (FPGA) for sensor control, data acquisition, and direct digital-to-analog control of the optical scanning system, making the application of the technology robust and efficient.

We report here a full-spectral fluorescence lifetime imaging (FS-FLIM) system capable of producing a full emission spectrum, time resolved at each wavelength for every image pixel. Our achromatic, confocal laser scanning, high spectral resolution, fluorescence lifetime imaging system is enabled by a 512-channel time correlated single photon counting (TCSPC) sensor capable of "on-chip histogramming" in each spectral channel[21]. Each channel of the sensor consists of 16 single photon avalanche diode (SPAD) detectors in two columns, each 95.12 μm in height. The system presented offers a transformative approach to FS-FLIM with unprecedented speed, spectral resolution, and versatility with configuration, in real-time, of spatial, spectral and temporal ranges allowing straightforward tuning of these parameters to specific experiments. Image acquisition was possible within 6 s with unprecedented full spectral and temporal resolution (512 wavelength channels × 32 time-bins × 256 × 256 pixels, with exposure times of 85 μs/pixel). The ultimate frame rate is, like all optical imaging techniques, limited by the overall photon budget available, set by the brightness and stability of the sample in question. This communication focuses on the depth of fluorescence lifetime information obtained with the system from a range of samples including freshly resected human lung tissue. Full spectral and temporal resolution lifetime-imaging is demonstrated with diffraction limited performance, 50 ps time resolution and 0.5 nm spectral resolution across the visible spectrum. These samples highlight the potential of true FS-FLIM to interrogate complex organic structures and demonstrate the variety of optical fingerprints exhibited by sample fluorophores.

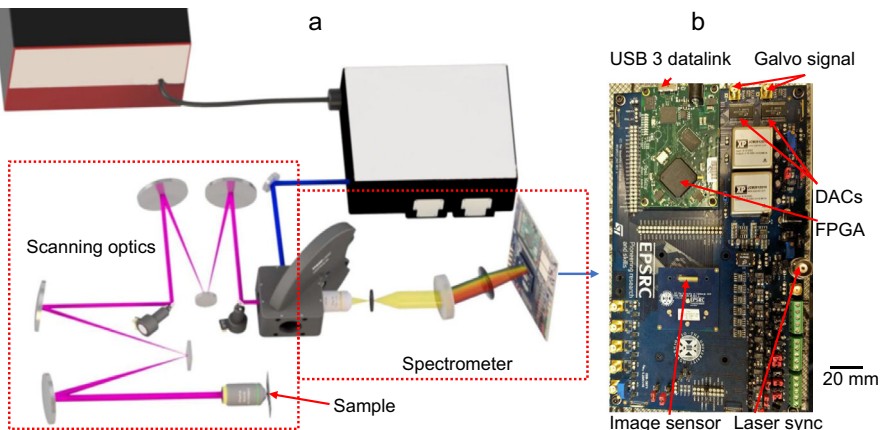

**Fig. 1 Optical imaging system layout. a** Optical path of the system (not to scale). The incoming supercontinuum laser is filtered via an acousto-optic filter allowing up to 8 colors to be selected before being directed to a filter wheel where a dichroic reflects the beam into the scanning optics. Two galvo mirrors scan in X and Y to form a stationary point onto the back plane of the primary objective. Returning fluorescence is de-scanned and passed to the spectrometer via a pinhole where it is dispersed onto the image sensor. Further details of the optical setup can be found in the methods section. **b** The imaging line sensor mounted on a printed circuit board (PCB) along with the field programmable gate array (FPGA) and two digital-to-analog converters (DACs) which are used together to control the position of the galvo mirrors. The scale bar relates to the PCB, not to the rest of the optical system shown in **a**.

## Results and discussion

**The FS-FLIM imaging system**. The system is equipped with a custom image line sensor[21] with 512 spectral channels with sensitivity between 450 and 900 nm. To achieve optimal performance over such a wide wavelength range it was important to minimize the chromatic aberration common in optical systems that employ transmissive optics. We thus designed a fully reflective optical path, except for the final objective, as high-end microscope objectives are designed to have inherently low chromatic aberration. This approach provided an optical path for diffraction limited performance from 400–900 nm. The highly efficient optical system had an overall transmission of >65%, which coupled with the sensitive sensor (photon detection efficiency up to 17%[21]), enabled the extremes of the emission spectra to be captured, even in relatively photon-starved environments, such as intrinsic tissue fluorescence imaging.

Figure 1a shows the optical layout of the imaging system incorporating a supercontinuum white light laser filtered through a tunable acousto-optic filter. Two galvanometer mirrors (galvos) were used for beam scanning onto the primary imaging objective. The galvos were separated by re-imaging spherical mirrors forming image planes at the second galvo and back aperture of the primary imaging objective. Whilst these mirrors introduce a small level of astigmatism to the beam with a modeled peak to valley wavefront error over the entire pupil of 0.139 waves, this has little detrimental effect on the system's point spread function and greatly simplifies the optical alignment.

Returning fluorescence was de-scanned through the same optical path where light then passed through a secondary objective and pinhole before recollimation into the spectrometer. Fluorescence entering the spectrometer was spectrally dispersed by a volume phase holographic grating (600 lines per mm, blazed at 600 nm) onto the sensor. In the presented configuration the spectrometer covered a spectral range of 500–760 nm, which was uniformly distributed over the 512-pixel sensor giving a resolution of ~0.5 nm with simultaneous lifetime measurements possible in each 0.5 nm band (these exact parameters can be tuned to specific applications, requiring different detection ranges, through alteration of the grating line number and focusing lenses). Full system details are given in the methods section. An important feature of the optical system is that detection of the emission is completely filter-less, with the entire spectrum captured for each pixel in a single image exposure, other than the dichroic filters required to remove the excitation laser lines. The PCB holding the sensor (Fig. 1b) hosts the FPGA together with two digital-to-analog converters that control the galvos, enabling easy synchronization of the scan with the sensor control and data acquisition functions. The system was controlled using custom Matlab 2019b scripts which performed final image assembly, processing, and display. Post-processing allowed for further interrogation of the information contained in the FS-FLIM data cubes.

**FS-FLIM of *Convallaria majalis***. Figure 2 shows a 512 spectral channel FS-FLIM image from directly observed *Convallaria majalis* labeled with Safranin and Fast Green fluorophores, using an exposure of 500 µs per pixel to ensure sufficient photon collection from the extremes of the emission spectrum. The sample was excited at 475 nm and an image of 256 × 256 pixels was captured. Specifically, Fig. 2a–d shows the spectral intensity information presented via color mapping, where, in post-processing, each spectral channel was assigned RGB color values according to its wavelength (Fig. 2a–c) and the contributions from each 0.5 nm spectral "band" summed to produce a single set of RGB color intensity value per image pixel. A full-color image

(Fig. 2d) was produced by combining the RGB color channels. A standard intensity weighted transparency was used to modulate pixel saturation (see methods). Figure 2e shows spectral lifetime histograms for two individual pixels, showing clear differences in spectral and temporal response corresponding to the different fluorophores, Fast Green (650 nm image) and Safranin (550 nm image).

In order to maximize the speed of imaging and minimize the light intensity directed onto the sample there is a desire to use as low a photon count as possible, concomitant with sufficient accuracy in the measured lifetime. With our high number of simultaneous spectral channels, a 256 × 256 pixel image can require over 33 million lifetime fits and so the computational load of the fitting method is an important consideration. Least squares fitting was chosen to process lifetime calculations for the presented study due to speed of computation. The trade-offs between this method and other analysis techniques are well covered in the literature[22]. After the dark count rate of the sensor has been subtracted a threshold is applied based on the collected intensity in a specific sensor channel. The threshold for lifetime calculations was set at 10 times the background, i.e., fluorescence events in an image pixel are 10 times the background noise resulting from the detector dark count and any scattering within the optical system. For any sensor channel with any image pixel where the signal was below this threshold a lifetime calculation was not performed and the pixel value set to zero.

Figure 2f shows two slices through the spectral lifetime data cube at 650 nm (top) and 550 nm (bottom). Since lifetime images are often very structurally flat, due to a lifetime decay being relatively independent of its intensity, the transparency of the lifetime images were weighted by the intensity of the fluorescence image to increase visibility of structurally related detail (a common practice that does not affect color scales[23,24]). From the lifetime images it is clear there is large variation in lifetime within regions of similar emission wavelength and conversely spectrally distinct regions with similar lifetimes. This shows the complex interaction of the two labels with the tissue and inter-dye interactions which were exposed using the combination of spectral lifetime information. A video walking through the whole FS-FLIM data cube demonstrating how the lifetime contributions vary across wavelengths is shown in the supplementary information (Movie 1—Full Spectral FLIM of Convallaria).

**FS-FLIM of a honeybee wing**. Figure 3 shows 512 spectral channel FS-FLIM data taken from a non-labeled, fixed honeybee wing. Figure 3a shows a heat map intensity image (256 × 256 pixels) summed over all spectral channels (left), a color image, showing variation in emission wavelength (normalized, middle) and the full wavelength spectrum of two regions of interest (right). Figure 3b explores the lifetime variation across the spectral emission with lifetime images displayed from 8 wavelength bands of 1 nm width selected from the full data cube (a video of the whole 512 channel FS-FLIM cube can be found in the supplementary information, Movie 2—Full Spectral FLIM of a Honeybee Wing). In Fig. 3a, the autofluorescence spectral emission shows subtle variations between pixels, with small changes of only a few nanometers apparent, which would be missed with traditional spectral imaging systems with significantly fewer spectral channels. In addition to the spectral changes in intensity there is significant spatial lifetime variation across the sample, linked to a strong spectral dependence, as indicated by the complex histograms shown in the bottom row of Fig. 3b. Each histogram corresponds to the distribution of lifetimes in the whole image at a specific wavelength, with lifetime values calculated by least squares fitting to the decays[25]. The

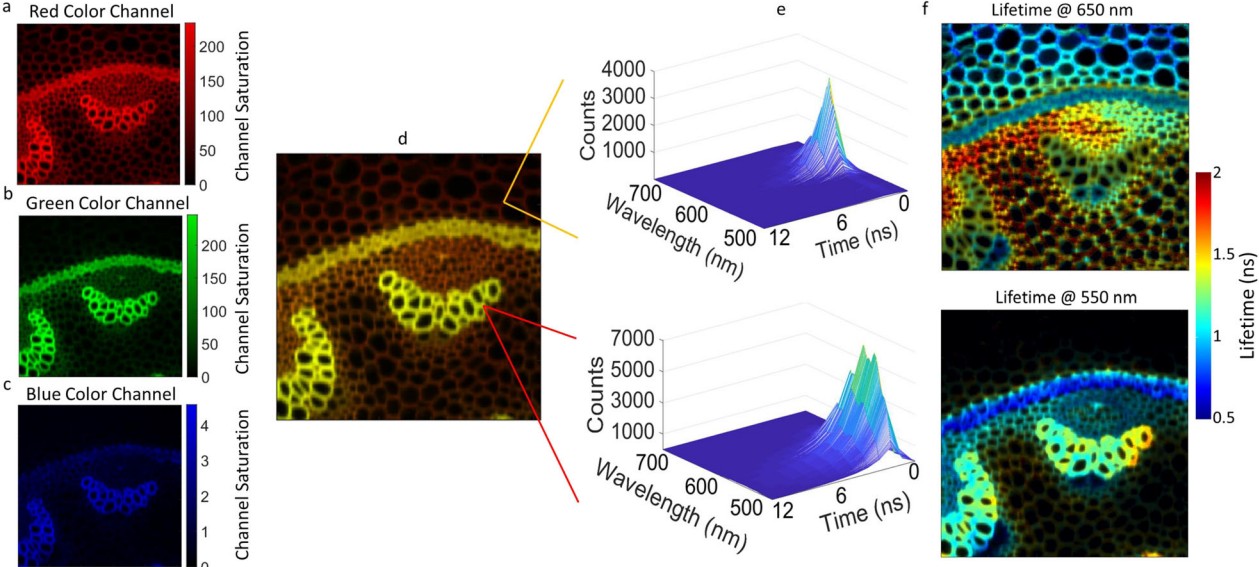

**Fig. 2 FS-FLIM of *Convallaria majalis* and spectral color mapping. a–c** Color channels assigned from the sensor spectral channels. **d** The combined RGB pixel data producing a color image (256 × 256 pixels with an image size of 600 × 600 μm). **e** Two contrasting pixels selected and their full spectral lifetime data cubes revealing the large variation in FS-FLIM data across the image with an area dominated by Fast Green (650 nm) and the Safranin (550 nm). **f** Two slices through the data cubes showing lifetime at two different wavelengths and showing variation in lifetime across the sample. The sample was imaged 3 times to insure repeatability.

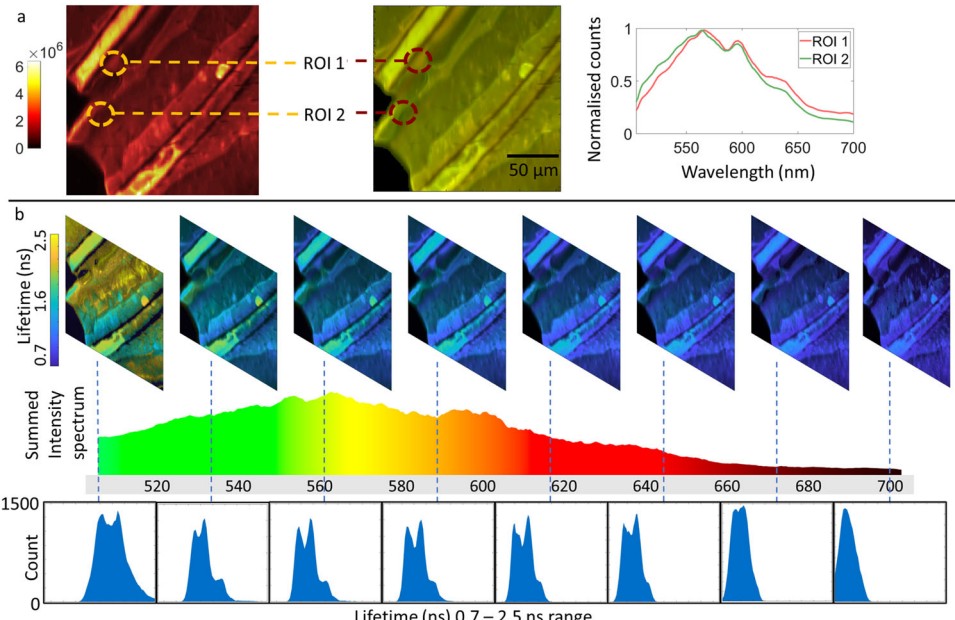

**Fig. 3 Full spectral fluorescence lifetime imaging of a fixed honeybee wing. a** Intensity image summed from the whole FS-FLIM data cube, a color image of the intensity data produced as per Fig. 2 showing normalized spectral variations across the sample along with normalized spectra from two regions of interest (ROI) within the sample highlighting the subtle spectral variations. **b** Slices through the FS-FLIM cube at eight wavelengths (top), the emission spectrum for the whole sample colored by emission wavelength, and the lifetime histograms for each presented wavelength. An exposure time of 500 μs per pixel was used to provide sufficient signal to noise ratio for the majority of the emission spectrum. Where the signal was reduced at the wavelength limits of the spectrum a moving spectral averaging filter was applied of a width of 8 pixels (~4 nm) to insure an adequate signal to noise ratio of >10. Whilst the use of such a moving filter does impact on the overall spectral resolution on these areas, far more information is retained than using static spectral binning as is enforced by systems with fewer spectral collection channels. The sample was imaged 3 times to insure repeatability.

histograms show multiple peaks across the spectrum strongly indicating different emitting species or environmental factors. The most likely explanation for the observed spectral and lifetime changes across the sample being areas of increased stiffness that have been shown to red-shift the autofluorescence emission wavelength[26] due to increased sclerotization (increased cross-

linking of proteins[27]). The lifetime reduction is most likely caused by increasing rates of non-radiative energy loss through self-quenching and local energy transfer to the cross-linked protein matrix. The combination of lifetime and spectral information allows for the subtle changes in the protein matrix to be observed. Whilst a detailed analysis of the underlying biological makeup of

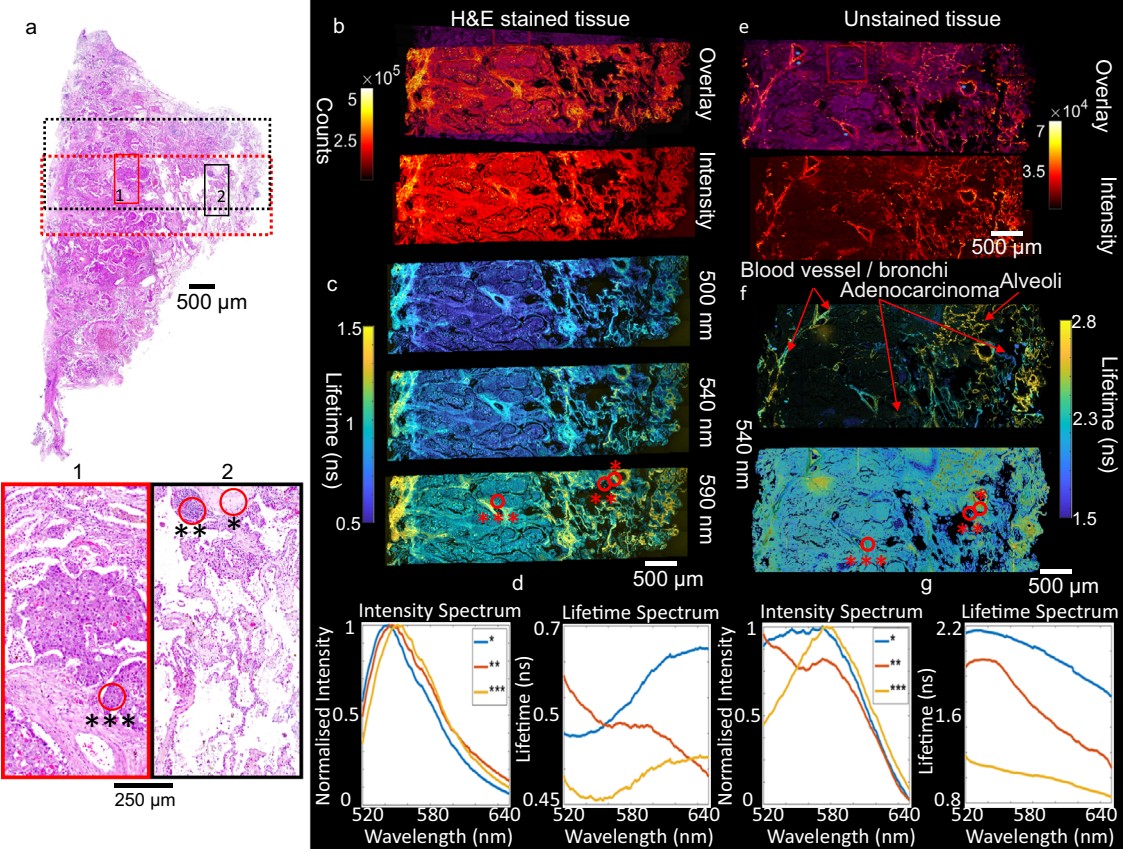

**Fig. 4 Full spectral fluorescence lifetime imaging of a histology slice of lung tissue. a** Haemotoxylin and Eosin (H&E) stained histology image of lung tissue with increasing cancerous (adenocarcinoma) tissue (from right to left). Highlighted areas show cancerous (1) and visibly healthier (2) regions. Dashed areas indicate the regions imaged for the stained (red) and the equivalent unstained (black) slice. Three regions of interest are highlighted. **b** Intensity based imaging of a stained histology slice both with and without overlay of the histology image. Tiled images consist of 4 rows of 12, 256 × 256 pixel images with a field of view of 600 × 600 μm each which were taken with 1 ms exposure time per pixel. A series of images were taken and subsequently combined to form a composite image showing the complete area (~6 × 2 mm). **c** Corresponding lifetime images at different wavelengths. **d** Mean spectral intensity (left) and spectral lifetime (right) for the three regions of interest. **e** Intensity imaging of an unstained slice both with (top) and without (bottom) overlay with the histology image. Images consist of 5 rows of 12, 256 × 256 pixel images, each ~600 × 600 μm, 1 ms exposure time per pixel. **f** Lifetime image at 540 nm displayed with intensity-based color saturation and an inverted color saturation. **g** Mean spectral intensity and spectral lifetime or the three regions of interest. The data shown is from a single sample per slice with 60 FS-FLIM images per slice, taken once each, one histology image was taken.

the sample is not the purpose of this work it can be clearly seen that full spectral lifetime imaging provides an exceptionally powerful tool for the study of the complex composition of biological systems.

**FS-FLIM of fixed human lung for histology**. We then applied 512 channel FS-FLIM direct imaging to spectral histopathology, an area of intense interest[15–18], offering the promise of rapid machine learning-assisted analysis[28] of clinically significant samples. The data shown in Figs. 4 and 5 originate from a freshly resected sample of human lung adenocarcinoma. The sample was both directly imaged as a fresh tissue sample (Fig. 5) along with preparation of two 4 μm slices as used for standard histology (Fig. 4). The slices were taken from the same area of the sample as measured on the fresh specimen with one being Hematoxylin and Eosin (H&E) stained and the other left unstained.

The H&E image (Fig. 4a) shows the transition from cancerous (example shown in Fig. 4a, 1) to increasingly mixed cancer and healthy tissue (transitionary) (example shown in Fig. 4a, 2) as determined by an independent histologist. The intensity-based images, shown with and without overlaying of the H&E image

(Fig. 4b, e), demonstrate good spatial agreement, with both samples showing increased structure in the transitionary region. The FS-FLIM data for the stained slice (Fig. 4c), which is dominated by the strongly fluorescent Eosin stain, shows a clear reduction in lifetime across the spectrum in cancerous regions.

Three regions of interest were selected containing differing cellular populations, as observed in the standard H&E image, are highlighted in Fig. 4a, 1 and 2. The spectral lifetime response of Eosin is shown for these regions of interest, which are shown co-located in the FS-FLIM image in Fig. 4c. Whilst the intensity spectrum of the emission from each of the regions is of similar shape with only a small shift in the location of the spectral peak (Fig. 4d, left) there is far greater contrast in the spectral lifetime response. The shift in Eosin lifetime is likely to be due to a multitude of factors including variation in cellular uptake, with high concentrations leading to common lifetime effects such as self-quenching and inter/intra cellular variations affecting local pH and viscosity. Different pH environments have been shown to affect the emission properties of Eosin such as a red-shift in the spectra, as observed here, and reduction in emission intensity, commonly associated with quenching, resulting in reduced lifetime[29]. Furthermore the presence of the hematoxylin stain

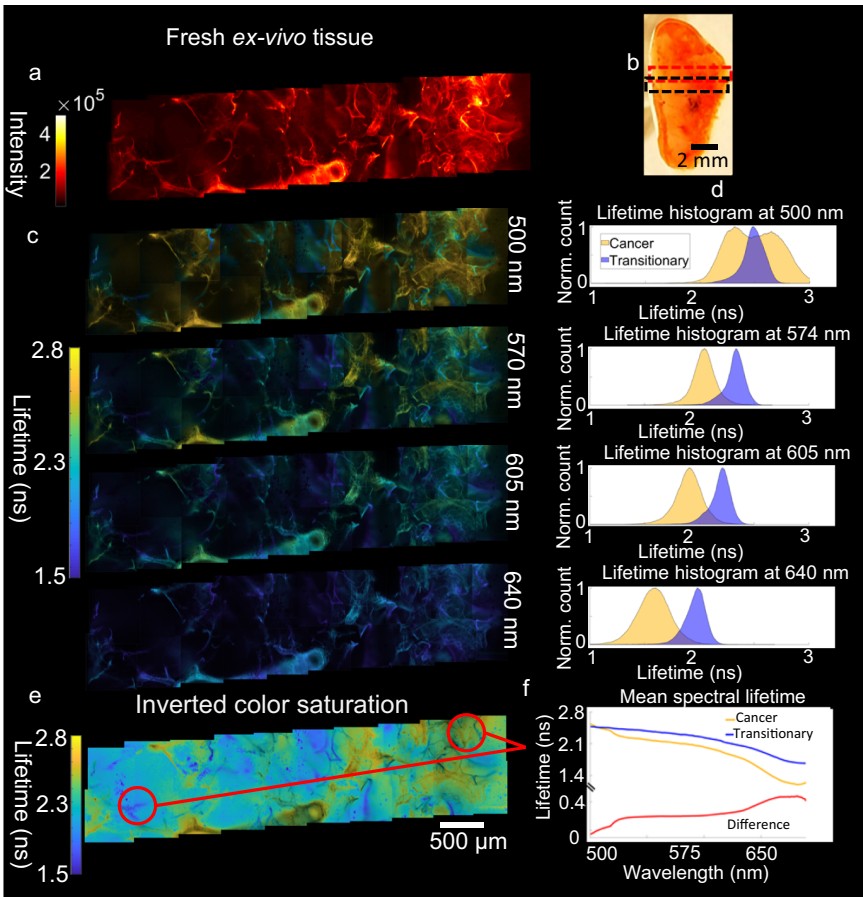

**Fig. 5 Full spectral lifetime imaging of a fresh, ex-vivo lung tissue sample. a** Intensity based image of the fresh ex-vivo tissue sample from the same surface used to gather histology slices (Fig. 4) with increasing structural detail from left (cancer) to right. **b** An image of the tissue sample as measured, with the dashed box indicating the section of the sample from which imaging was acquired (2 × 10 images at 256 × 256 pixels and ~600 × 600 μm covering an area of ~6 × 1 mm). **c** A series of lifetime images taken from the FS-FLIM data cube at various wavelengths showing the progression of lifetime through the cube. **d** Normalized (Norm.) lifetime histograms corresponding to two regions of interest, cancerous and transitional, for the wavelengths shown in **c**. **e** A set of images from 570 nm with an inverted transparency channel, highlighting cancerous regions. **f** Mean spectral lifetime from the equivalent regions of interest shown (cancer, left, transitory region, right) along with the difference between them. Due to the constraint of sample availability the fresh tissue sample was imaged once for each data cube location, and consistency across the 20 images shows repeatability of the technique. Note: this publication does not aim to show the cross-sample significance of the lifetimes shown.

has been shown to increase the observed Eosin lifetime where co-staining of cells occurs[22].

Whilst the stained slice data shows the potential power to distinguish regions of interest using FS-FLIM, of more utility is the use of unstained tissue (Fig. 4e, f). From the lifetime images shown in Fig. 4f there are pronounced differences in lifetime between amorphous cancerous regions (left) and healthy alveolar (right). The labeled regions of Fig. 4f are representative areas of different lung structure as determined by a lung histologist. It is clearly seen that whilst there is a general trend to longer lifetime towards the right of the image, which contains healthy tissue, the sample remains heterogeneous with large variations in lifetime. To increase visual contrast in the lower intensity regions the transparency channel based on intensity was inverted (Fig. 4f) which clearly shows the greater amorphous structure of tissue that is typical of cancer. The same three areas of interest as before are shown co-located in Fig. 4f showing significant spectral intensity differences between the regions, with region three (***) located in the most cancerous area exhibiting a red-shifted emission consistent with previous reports[12]. The spectral lifetime also shows distinctly different responses from the three regions with a reduction in lifetime observed as cellular density and increasing levels of amorphous tissue normally associated with

cancer manifest[14–17]. The collection of both spectral and lifetime information that the presented system enables, with high levels of detail, allows for increasingly robust distinction between cellular regions with some species showing large spectral variation with little lifetime change, and others showing distinct changes in lifetime for a simpler spectral response.

**FS-FLIM of human lung tissue**. Figure 5 shows FS-FLIM images of the sample of fresh, ex-vivo, lung tissue that was analyzed in Fig. 4, before histological processing. Figure 5a shows intensity images taken over the region highlighted in Fig. 5b. As with the sliced sample, increasing structure is observed in the healthier region and a large airway is present in the lower portion of the images. The complex nature of the sample fluorescence is fully revealed in the FS-FLIM images (Fig. 5c), where four selected wavelengths (together with lifetime histograms (Fig. 5d) taken from an area of suspected cancer are shown along with a highly structured region within the healthier region. Multiple emitting populations are observed at shorter wavelengths showing that neither region is wholly of one type. This is expected as variation in cellular makeup across the specimen was observed in the sliced samples, and again shows the power of spectral lifetime imaging, that offers contrast not seen in the standard intensity image.

Similar to the regions of interest in the tissue slice data, there is a reduction in lifetime in the more amorphous cancerous regions. It should be noted that the regions of interest analysed here are of significant size, at approximately 2,500 μm² and so are likely to include multiple cell types. The mean spectral lifetimes, Fig. 5f, from the two areas, and the difference between them, highlights the complex spectral lifetime dependence and further illustrates the potential for forming a FS-FLIM fingerprint for distinguishing the tissue types in fresh, unprocessed, samples with the ultimate application of in-vivo real-time pathology.

## Discussion

The goal of this work was to demonstrate the underlying capabilities of the presented system to acquire an increased level of spectral and temporal information. Whilst image capture and histogram formation occur in real-time there is a bottleneck to sequential imaging speed due to data transfer of the spectral data from the FPGA to the PC (currently via USB 3 connectivity). The theoretical maximum size for a 256 × 256 pixel image captured with 512 spectral and 16 (chained) temporal bins is 16 Gb. This can lead to a delay of up to 30 s in the worst-case scenario in displaying the image. In practice the image size is dependent on the density of sample in the image and the breadth of the emission spectra involved, reducing the image size and associated transfer times. There is potential to reduce this data retrieval time by optimizing of the data link to the PC to enable real time imaging at 0.2 frames per second. As presented in Figs. 1 and 2, a high spectral contrast color image can be obtained through three channel (typically RGB) color-mapping requiring only 3 data points per pixel. Since this can be performed on the FPGA before transmission to the PC it represents a possible solution to maintaining spectral contrast whilst requiring low data flow. Clearly if full spectral lifetime information is required the entire, 512 spectral channel, time resolved, dataset must be transmitted to the PC. Here the inherent flexibility in the design of the core architecture comes to the fore allowing spectral and temporal binning to be applied through minor changes in software choices to increase frame rates before transmission to the PC. This enables imaging at up to 10 frames per second (at 128 × 128 pixels). The application of the system to high frame rate imaging utilizing dimensional binning is the scope of future work. The approach of pixel rejection, or application of a moving filter of 8 pixels (4 nm), was used to ensure a signal of >10 background for lifetime calculations. This was applied after correction of signals through subtracting sensor dark count and sensor spectral efficiency response. This approach ensured sufficient photon events in the resulting decays for fitting errors to be dominated by Poisson noise; however, it should be noted that the fitting model used assumes a single exponential decay. Clearly in regions with perturbed emission or multiple emitting species this will introduce deviation from true lifetime values; however, this can be overcome by a more complex fitting approach at the sacrifice of computational speed. For fresh tissue samples, there is potentially some depth penetration of excitation light and therefore collection from multiple species; however, the diffraction limited confocal volume probed is small relative to structures of interest and is not deemed to have a significant perturbation on the signals collected. Similarly, the de-scanned confocal nature of the system causes minimal disruption to the spectral signal as the spectrometer is effectively decoupled from the image scan, and the optical path-length differences due to axial penetration are too small to influence the optical timing response.

The detailed analysis of the underlying biological processes that give rise to the optical signatures presented is not the aim of this work. However, the power of FS-FLIM to fingerprint different species is clear, and a detailed investigation of individual cellular properties is for future investigation. Optically fingerprinting cellular types has the potential to streamline the pathology pathway by removing the need for tissue staining and the associated preparation work. Whilst it is not the intention of this study to definitively define the distinctions of cancerous and non-cancerous tissue, we demonstrate here the contrast possible using our powerful FS-FLIM technology, illustrating its promising potential route to high-throughput histopathology applications that remove the need for sample processing and fixing.

In summary we have demonstrated a highly sensitive, versatile, and robust full spectral fluorescence lifetime imaging microscopy system that allows for rapid data acquisition from a wide range of sample types. Capture of the entire fluorescence lifetime spectral data cube will enable multiple applications such as full spectral lifetime Förster resonant energy transfer (FRET) imaging, simultaneous fluorescence and Raman imaging and unprecedented adaptability for multi-fluorophore analysis, techniques that have applications throughout the life sciences.

## Methods

**Optical system**. The complete optical system is shown in Fig. 1. The incoming supercontinuum laser (NKT Evo HP, pulse duration <100 ps, repetition rate 20 MHz) is filtered using an opto-acoustic filter (NKT Super-K Select) allowing up to 8 spectral lines between 400 and 700 nm each with a bandwidth of ~2 nm and around 2 mW average power. The collimated output of the filter unit was expanded to a diameter of 3 mm to match the subsequent scanning optics. The beam was passed onto the first scanning mirror (Thorlabs GVS202) before being re-imaged onto the second scanning mirror via two 50 mm diameter, 300 mm radius of curvature mirrors (Thorlabs CM508-150-E02) and a plane 25 mm diameter fold mirror (Thorlabs BB1-E02). The second scanning mirror was subsequently re-imaged using a similar configuration with a pair of 50 mm diameter spherical mirrors of 300 mm and 400 mm radius (Thorlabs CM508-200-E02) of curvature and a fold mirror which introduced a magnification of 1.5 times to ensure the back aperture of the objective was filled. All mirrors were standard silica with a dielectric coating providing reflectance >99% between 400 and 900 nm. This achromatic scanning system produced an XY scan at an image plane at the back aperture of the primary objective (Olympus 20 × 0.5 NA plan fluorite air objective used for all samples except for the honeybee wing which used an Olympus 60 × 1.25 NA plan fluorite oil immersion objective). Fluorescence returning from the sample was "de-scanned" through the same optical path, before being separated from the excitation light by a conventional dichromatic filter set (Semrock 488 nm Brightline long pass). The dichroic filters were mounted in a 5 filter wheel allowing rapid adjustment of cut-on wavelength. Whilst the rest of the optical path is filterless the choice of laser reflecting dichroic is important to minimize omissions in received emission spectrum whilst allowing the use of the tunable excitation source. The fluorescence subsequently passed through a 10 × 0.25 NA objective (Olympus plan fluorite) focused onto a ~100 μm iris which acted as system pinhole before it was collimated via an achromatic doublet lens (focal length 60 mm—ThorLabs AC508-075-A), directed onto a transmissive holographic grating with 600 lines/mm (Wasatch Photonics WP-600/600-50.8), followed by focusing through a second achromatic doublet (focal length 60 mm—ThorLabs AC508-075-A) to produce a line on the sensor matched to the sensor pixel height of 95.12 μm.

### Sample preparation and image processing

Convallaria majalis. For direct observation, via a 20 × 0.5 NA objective (Olympus), the Convallaria sample (slice labeled with Fast Green and Safranin, Leica) (Fig. 2a–d) was excited at 485 nm with emission separated via a 488 nm dichromatic filter (Semrock 488 nm Brightline long pass) to capture the complete emission spectrum. An image was captured using a 500 μs pixel exposure time and the complete set of 512 spectral and 16 time channels (the time channels were used in "chained mode" whereby two channels are "chained" together to allow increased dynamic range[21], see reference for more detail) were passed to the PC for processing where color information was processed as described in the main text. Lifetime values were calculated by using a rapid least-squares fitting algorithm, implemented using the open-source GPUfit library[25] in Matlab 2019b (Mathworks).

*Honeybee wing.* For direct observation, via a 60 × 1.2 NA oil objective (Olympus), of the Honeybee wing (fixed, Brunel Ltd) shown in Fig. 3, the same settings were used as for the Convallaria sample with 512 spectral and 16 time channels. Color RGB values for color images were processed as described in the main text. After RGB values were obtained, the saturation of each pixel was adjusted based on the corresponding overall intensity for that pixel. This was performed by setting the image background to black and adjusting the pixel transparency value, scaled by its

normalized intensity. Lifetime images were selected across the wavelength spectrum and were similarly processed with a transparency weighted by the fluorescence intensity at the same wavelength. Lifetime decays were again fitted using the least squares method utilizing GPUfit.

*Ex-vivo human lung tissue section and tissue slices.* The ex-vivo human lung tissue sample, used for 512 spectral and 16 time channels imaging in Figs. 4 and 5, was obtained from a patient with non-small cell lung cancer (measuring >3 cm in diameter on pre-operative chest computed tomography) undergoing thoracic resection surgery (see ethics statement below). The whole tissue section was approximately 6 × 12 mm (taken to encompass regions of both frank cancer and the margin towards healthy tissue).

To image the fresh tissue (Fig. 4), the sample was placed onto a 35 mm Fluorodish (ThermoFisher) and onto the sample translation stage of the microscope. Directly observed images were recorded with a 0.5 NA 20x objective (Olympus) at 256 × 256 pixels covering an area of ~600 × 600 μm each with a pixel dwell time of 1 ms, which was chosen to provide sufficient light at the extremes of the tissue fluorescence decay and in each spectral channel. All other settings were the same as for the previous directly viewed samples. Each image was taken sequentially (2 rows of 10 images) with the translation stage moved 0.5 mm per image to provide ~50 μm overlap between images for stitching. At the edges of emission spectra where counts are lower, noise in the lifetime-based images was reduced by using a moving spectral mean of 8 channels width (~4.5 nm) across the spectral region for lifetime calculations to increase signal to noise. Lifetime calculations were performed using least squares fitting as described above. Furthermore, a 200 count threshold was applied, which equated to a signal to noise ratio of ~10:1, below which the lifetime data was discarded. The intensity-based images shown are produced by summing over the whole 512 spectral channels and 16 time channels. The color map was then normalized across the whole set of images to the brightest pixel to provide a true, sample wide, representation of intensity variation. As for the Honeybee wing, lifetime images were processed with a transparency weighted by the fluorescence intensity at the corresponding wavelength. The intensity weighted transparency has the effect of bringing out areas of lifetime corresponding to higher intensity. A set of images with the intensity weighting inverted was also used to bring out low count regions of the image.

The same ex-vivo tissue specimen was fixed in 4% neutral buffered formaldehyde, embedded in wax with 4 μm slices cut onto slides (taken from the same sample plane as measured in Fig. 5). Sequential samples were then used as unstained or stained with Haematoxyln and Eosin and co-registered (AxioScan.Z1, Zeiss, Germany). The slides were imaged directly under the same conditions as the fresh ex-vivo tissue sample.

**Ethics**. All experiments using ex vivo human lung tissue were performed following a favorable ethical opinion received from the South East of Scotland Research Ethics Service REC 1 (on behalf of the National Health Service), and approved by the NHS Lothian NRS BioResource (REC ref: 13/ES/0126 and 15/ES/0094). All subjects gave written informed consent.

**Reporting summary**. Further information on research design is available in the Nature Research Reporting Summary linked to this article.

## Data availability
Data underling Figs. 2–5 generated in this study have been deposited in the University of Edinburgh DataShare facility under accession code https://doi.org/10.7488/ds/3099.

## Code availability
Custom Matlab scripts and associated open-source code for data analysis are available through the University of Edinburgh DataShare facility under accession code https://doi.org/10.7488/ds/3099.

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

## Acknowledgements

We would like to thank the Engineering and Physical Sciences Research Council (EPSRC, United Kingdom) Interdisciplinary Research Collaboration (grant number EP/K03197X/1 and EP/R005257/1) for funding this work. We thank Fiona Kerry for their initial work on the design of the device.

## Author contributions

M.B., J.M.G., and R.K.H. developed the original ideas. G.O.S.W., E.W., and J.M.G. designed and built the scanning system. R.K.H. and A.T.E. developed the line sensor.

A.T.E., G.O.S.W., and N.F. characterized the sensor. G.O.S.W., E.W., A.T.E., and N.F. characterized the scanning system. A.T.E. wrote the sensor firmware. N.F., A.T.E., Q.W., and G.O.S.W. developed the software and data analytics. Q.W. wrote the GPU acceleration code. G.O.S.W., E.W., S.F., and N.F. performed the experiments. S.F., A.R.A., and K.D. supplied the samples and performed clinical analysis. M.B., J.M.G., R.K.H., and K.D. supervised the project. G.O.S.W., E.W., N.F., M.B., J.M.G., and R.K.H wrote the paper. All authors revised the paper.

## Competing interests

The authors declare no competing interests.
