## [Peer Review File · Nature Communications]

Reviewers' comments:

Reviewer #1 (Remarks to the Author):

In this manuscript, the authors present an exciting development in multi-spectral lifetime imaging, but they do so in such a manner that it is really not possible to digest what they have done, even after reading it multiple times. The claims of spectral resolution, lifetime resolution and framerates are very challenging to understand, and are almost misleading. For example, the authors claim a spectral resolution that is impressive in their title, but they present data with many of the channels averaged together (16 at a time) which reduces the resolution to about the same as the commercial units available. Furthermore, for many of the studies, it appears that they binned the wavelengths into only two bins. This is far less than the commercial units available from many vendors. In other images, very low spatial resolution was used (less than 200 x 200 pixels).

I am excited by the work and the impact it might have on the field, but it will only have an impact if it is clearly presented.

This could be an amazing contribution, but it must be made much honest with respect to the imaging conditions. It is fine to point out that the spectral, lifetime, framerate and spatial resolution represents compromises, and be honest that it is not possible to have the best of all of them. I encourage a more transparent presentation of the amazing capabilities that this instrument can offer.

Reviewer #2 (Remarks to the Author):

The authors present in the manuscript their optical set-up, a fully achromatic confocal laser scanning system equipped with a maximum of 512 spectral channels and 32 time channels sensor. The system is characterized by 50 ps time resolution and 0.5 nm spectral resolution.

The authors also exploit their system to characterize the spectral and temporal properties of multiple samples, from the standard *Convallaria majalis* to ex-vivo and fixed human lung tumor tissues.

Although the manuscript is interesting and well written, in my opinion the authors did not provide data and applications that can be considered a real breakthrough.

In particular, I would raise the following points:

- their novel optical system is very poorly described. The authors did not provide any technical details about the performance of their optical system and did not compare it to any of the already implemented (commercially available or not) in the literature so it is very difficult to understand the real novelty they claim. Moreover, in the initial part of the manuscript, the biological/physical/optical problems they would solve are not properly contextualized.

- the range of applications reported did not highlight the potentiality of their system. In my opinion, it is not deeply discussed and novel extensive and in-depth conclusions about the experiments are not reported.

In my opinion a lot of points remain unanswered:

- Which is the importance of spectral resolution in their applications? Which is the difference in the obtained results by operating a different binning of the spectral channels?

It would have been interesting a discussion of the improvement of the full spectral method with 0.5 nm resolution with respect to the hyperspectral imaging in terms of errors in the reconstructed spectra/acquisition time/acquired counts/information extracted from the images.

- Does the bleaching affect their images (both intensity and lifetime maps) when acquired with the maximum spatial and temporal resolution? Which is the effect of the z depth into tissue on the spectral and lifetime images? Are their 0.5 nm spectral and 50 ps temporal resolution affected?

Moreover, the authors reported only static images but it would have been interesting to show also if their system is able to characterize and follow dynamic events. It is important in my opinion at least a discussion about the eventual limitation in the number of spectral channels that can be exploited and the errors in the computed lifetime in dependence on the temporal event under investigation.

- How many counts they used to compute the lifetime? For example, in Figure 2g (and related video) the authors report a count scale in the range [0-150 counts]. Are these counts exploited to extract the lifetime map in Figure 2h? Which is the error associated with each lifetime due to these few counts? Moreover, could the authors explain why there are some black pixels (0 ns lifetime) in the lifetime map associated with non-zero counts pixels in the intensity map and viceversa?

-Regarding Figure 3: I think the authors show a nice application, especially since they exploit the autofluorescence signal of the sample. However, I think that more information could have been inferred from the data. For example, from the spectral data is it possible to retrieve any differences among pixels by mapping the ratio between the red and green part of the spectrum (above-to-below 600 nm) instead of the mean wavelength?

Are the differences in the lifetime distributions due to different autofluorescent proteins in the samples or to a single protein exposed to different environments?

The authors should at least quantify the differences in the sample characterization that can be obtained by exploiting the spectral method, the lifetime imaging or their combination. Moreover, the authors did not report the errors neither the number of samples analyzed.

In my opinion it is not clear what improvements can be gained in the sample characterization obtained by means of their system with respect to other available spectral/lifetime setup. The authors should have stressed more the novelty of their system and the applications.

- In Figures 4 and 5 the authors did not report any error analysis in their graph nor the number of samples. Also in these applications, a more in depth analysis of the acquired data should have been reported. For example, is it possible to extract some interesting information by exploiting only the spectral images or the lifetime maps? Which is the improvement of using the combined information related to both techniques? And again, the improvement of their setup with respect to others should have been stressed more also in these applications. According to which parameter did the authors classify the cancer and healthy ROIs?

Are the differences in the spectral lifetime between cancer and healthy ROIs related to different cellular populations among these areas? A more refined comparison should be performed between the spectral/lifetime data and the H&E images. Moreover, is it possible to extract a parameter that allows to identify tumor areas among healthy regions in the acquired spectral/lifetime images and compare the results with those obtained by a pathologist?

Minor points:

-Figure 3b: the counts scale is missing in the lifetime histograms

-Lines 69-70: "A color image was produced by using an intensity weighted transparency alpha channel to modulate pixel saturation.". It is not clear (at least to me) the procedure followed by the authors to process the images.

Response to reviewers for the manuscript “Full Spectrum Fluorescence Lifetime Imaging with 0.5 nm Spectral and 50 ps Temporal Resolution”

We would like to start by thanking the reviewers for their comments and helping us to significantly improve the quality of the manuscript, enabling us to focus the reader on what are new advances in the field of spectral and lifetime imaging. To achieve this, we have undertaken a major revision to the paper structure to increase focus on the application of the system at full spectral and temporal resolution demonstrating its novel ability for simultaneous high spectral and high-resolution fluorescence lifetime acquisition. We have removed text and figure sections regarding temporal and spectral binning for increasing framerate and a comment regarding this has been added into the main text. To concentrate on the core advances which the reviewers agreed were significant, we have focused on the direct imaging removing the coherent fibre bundle applications. We have also made it clear that we are reporting a significant technological advancement and that the examples used are exactly that. Examples that the reader can quickly appreciate and then understand how the capability of our new instrument could help them to understand the complex, and very often sample specific biophysical effects taking place in terms of local lifetime or spectral changes.

Below we demonstrate how we have answered the specific points raised by each referee, indicating our response and examples of the changes in the text that have been made in blue. We believe we have answered all the points raised by the reviewers and explained our reasoning behind our response when required. Also included in a separate document named “Comparison_to_Previous_Submission” is a constructed comparison, produced in word 2000, between the initially submitted manuscript and the resubmission.

Reviewers' comments:

Reviewer #1 (Remarks to the Author):

In this manuscript, the authors present an exciting development in multi-spectral lifetime imaging, but they do so in such a manner that it is really not possible to digest what they have done, even after reading it multiple times. The claims of spectral resolution, lifetime resolution and framerates are very challenging to understand, and are almost misleading.

We totally understand the referee's comment and thus we have, as indicated above, made significant changes to the position of some of the text to ensure that the focused message comes through as indicated in our general comments above. Specifically, we have made the spectral and temporal performance clear, also indicating what is possible with improvements as we move forward. An example of the change is indicated below.

Example from text: “Here the inherent flexibility in the design of the core architecture comes to the fore allowing spectral and temporal binning to be applied through minor changes in software choices to increase frame rates before transmission to the PC. This

enables imaging at up to 10 frames per second (at 128 x 128 pixels). The application of the system to high frame rate imaging utilizing dimensional binning is the scope of future work”.

All presented examples now use full spectral imaging for clarity. We believe this greatly clarifies the message regarding our technology for the reader. Further we have expanded and clarified the text regarding the use of any spectral filtering for signal to noise considerations.

For example, the authors claim a spectral resolution that is impressive in their title, but they present data with many of the channels averaged together (16 at a time) which reduces the resolution to about the same as the commercial units available.

We think there has been a slight misunderstanding of how the data was collected and analysed, which is an indication again that we were confusing in our original text. No static binning was applied to the sensor data, all 512 channels were always acquired. The text has been updated for clarity and explanation added around the use of a moving, averaging, digital spectral filter. This was only applied when the signal was very low for the lifetime calculations to ensure signal to noise was sufficient for accurate fitting. This was typically applied at the edges of emission spectra. Such a moving filter has increased discrimination when compared to static binning of spectral ranges employed with fewer base channels, typically found in commercial systems. As mentioned above sections that did use higher levels of spectral/temporal binning, reducing the spectral resolution of the data collected to increase frame rates, for imaging through fibre bundles, have been removed to ensure there is no confusion.

Example from text:

“Where the signal was reduced at the wavelength limits of the spectrum a moving spectral averaging filter was applied of a width of 8 pixels (~4 nm) to insure an adequate signal to noise ratio of > 10. Whilst the use of such a moving filter does impact on the overall spectral resolution on these areas, far more information is retained than using static spectral binning as is enforced by systems with fewer spectral collection channels.”

Furthermore, for many of the studies, it appears that they binned the wavelengths into only two bins. This is far less than the commercial units available from many vendors. In other images, very low spatial resolution was used (less than 200 x 200 pixels).

As noted, we have removed figures and the majority of discussion around the use of temporal and/or spectral binning for increasing framerate other than the comment above. This focuses the version presented on novel full spectral and temporal resolution datasets.

I am excited by the work and the impact it might have on the field, but it will only have an impact if it is clearly presented.

This could be an amazing contribution, but it must be made much honest with respect to the imaging conditions. It is fine to point out that the spectral, lifetime, framerate and spatial resolution represents compromises, and be honest that it is not possible to have the best of all of them. I encourage a more transparent presentation of the amazing capabilities that this instrument can offer.

We thank the reviewer for the comments and believe the updates to the manuscript address these concerns.

Reviewer #2 (Remarks to the Author):

The authors present in the manuscript their optical set-up, a fully achromatic confocal laser scanning system equipped with a maximum of 512 spectral channels and 32 time channels sensor. The system is characterized by 50 ps time resolution and 0.5 nm spectral resolution. The authors also exploit their system to characterize the spectral and temporal properties of multiple samples, from the standard *Convallaria Majalis* to ex-vivo and fixed human lung tumour tissues.

Although the manuscript is interesting and well written, in my opinion the authors did not provide data and applications that can be considered a real breakthrough.

The goal of the manuscript was to present a new technology for acquiring time-resolved spectral information at a speed, resolution and ease of acquisition not previously possible. The samples imaged were selected as representative of the types of biological sample that could be explored with this very data rich approach. It was not the goal of the paper to undergo a detailed probing of biophysical processes underlying the signatures presented from the samples. Each figure is intended to show, in a different way, the volume of data produced and highlight initial areas where it provides for greater discrimination. We have updated the introduction bringing in comparisons with commercial and previously reported systems and highlighted where we believe we have made significant advances. This was perhaps not clear, as mentioned above, in the original submission. Our novel system has significantly more spectral detail, resolution, than those previously reported, with the potential for other spectroscopic imaging modalities such as Raman, in the presence of fluorescence. Furthermore, unlike most other reported systems, the level of integration presented provides a system that can be adopted rapidly without a high level of specialist knowledge. The data burden normally associated with time resolved imaging at this spectral resolution has also been a major bottleneck to such technologies. The integration of photon-counting electronics onto the sensor allows for a huge reduction in data flow and hence the ability to image at increased rates. Furthermore, as the system exhibits “optically filterless” detection, apart from the necessary notch dichroic to enable laser excitation lines to enter the optical path, the entire emission spectrum is always recorded and any desired spatial, temporal or spectral range adjustment or binning can happen, in real time, through software control. Such changes in data acquisition do not require any physical hardware changes or in the case of systems such as the Leica Stellaris require a complex and costly advanced optical filtering system.

In particular, I would raise the following points:

- their novel optical system is very poorly described. The authors did not provide any technical details about the performance of their optical system and did not compare it to any of the already implemented (commercially available or not) in the literature so it is very difficult to understand the real novelty they claim.

We have updated the introduction to address some specific comparisons with commercial and literature systems notably:

Example from text: “Some commercial lifetime imaging systems, such as the Leica Stellaris, can achieve increased spectral resolution through multiple rapid sequential image captures, with incremental shifts in detection wavelength. However, the requirement for multiple image acquisitions slows down the entire acquisition process and contributes greatly to detrimental effects such as photobleaching and loss in image quality due to sample motion. An optimal solution is to obtain the entire wavelength-lifetime spectrum for each image pixel in a single pass.”

Example from text: “Whilst previous work has utilized line arrays of SPADs^{4,20}, including the ability to time gate arriving photons on the sensor²¹, the implementation presented here offers two key advantages in addition to the very high number of spectral channels. Firstly, the ability to generate lifetime histograms “on-chip” greatly reduces the volume of data typically required for TCSPC lifetime imaging. This enables greater photon number processing throughput, resulting in increased signal to noise ratios, allowing for use of lower excitation light intensities and reduced imaging exposure times. Secondly, all of the photon timing electronics are contained on-chip removing the requirement for complex printed circuit board electronics and delay lines. The chip is packaged onto a single Printed Circuit Board (PCB) containing a Field Programmable Gate Array (FPGA) for sensor control, data acquisition, and direct digital-to-analog control of the optical scanning system, making the application of the technology robust and efficient. “

Example from text: “Capture of the entire fluorescence lifetime spectral data cube will enable multiple applications such as full spectral lifetime Förster resonant energy transfer (FRET) imaging, simultaneous fluorescence and Raman imaging and unprecedented adaptability for multi-fluorophore analysis, techniques that have applications throughout the life sciences.”

In addition, we have increased the methods section slightly to make sure that the full optical details are provided and removed any reference to coherent fibre bundle imaging which may have added confusion and been lacking in some specific details.

Moreover, in the initial part of the manuscript, the biological/physical/optical problems they would solve are not properly contextualized.

We have added some extra text to increase context, however, we believe we have stated clearly why other systems have been unable to achieve the level of time-resolved spectral detail presented, namely the data cost, alleviated by the chip level integration presented. Again, we believe that the other changes in the manuscript to focus on the essentials of our system help to address this point. Specifically we have added the following;

Example from text: “Unmixing of signals from complex fluorescent samples¹ is enhanced by high resolution time-resolved emission spectroscopy (TRES²), where the optically efficient, simultaneous acquisition of full emission spectrum and lifetime datasets enables maximal exploitation of the fluorescence signal and determination of small changes in emission profiles. The emission properties of individual fluorophores in complex environments such

as biological samples are affected by a host of inter-molecular interactions and environmental fluctuations including resonant energy transfer, pH, viscosity, temperature, and a range of quenching pathways that lead to shifts in both emission spectrum and lifetime of excited states³. When multiple endogenous and exogenous emitting species are present there are further complex interactions leading to subtle shifts in emission often only over a few nm.”

Example from text: “Full-spectral fluorescence lifetime imaging microscopy requires a volume and speed of data collection which has constrained previous systems to a small number of parallel spectral channels⁴⁻¹⁴. This has inherently limited the complexity of the signals that can be distinguished by such systems, reducing their ability to monitor subtle spectral or lifetime changes. Wide ranging influences can introduced slight perturbations in spectral signals, such as pH¹ or, in the case of tissue, structural or biomarker influences such as those found in cancer¹⁵⁻¹⁸.”

- the range of applications reported did not highlight the potentiality of their system. In my opinion, it is not deeply discussed and novel extensive and in-depth conclusions about the experiments are not reported.

As mentioned above the focus on this manuscript is the significant technological advance we have achieved. We do not believe that a detailed explanation of fundamental influences on molecular emission properties is appropriate for this manuscript, as the primary focus is on the technology not the photophysical understanding of the resulting optical signatures, which is a significant undertaking for future study. The figures and applications are intended to show the depth and richness of the data that can be acquired by such a system. We believe that the examples shown do demonstrate the potential to extract unprecedented optical signatures from relevant samples to a wide range of fields and as noted by the reviewer, the use of samples such as *Convallaria* is standard practice throughout the literature when demonstrating a new imaging technique. The subsequent interpretation of what the recorded details means would remove focus from the underlying technology and is the subject for specific publications on that particular sample type.

In my opinion a lot of points remain unanswered:

- Which is the importance of spectral resolution in their applications? Which is the difference in the obtained results by operating a different binning of the spectral channels?

As mentioned above the paper has been reworked in response to reviewer #1 to focus the discussion on full spectral, non-binned, datasets. We have added text around the types of signal that would not be observed with a system with less channels, for example.

Example from text: “In Figure 3(a), the autofluorescence spectral emission shows subtle variations between pixels, with small changes of only a few nanometers apparent, which would be missed with traditional spectral imaging systems with significantly fewer spectral channels.”

Where a moving digital spectral filter has been applied to edges of emission spectra with reduced signal this is now explained clearly, and as mentioned above, the use of such a (relatively narrow, 4 nm) filter results in significantly less data loss than the use of static filtering enforced by systems with fewer channels.

Example from text: “Where the signal was reduced at the wavelength limits of the spectrum a moving spectral averaging filter was applied of a width of 8 pixels (~4 nm) to insure an adequate signal to noise ratio of > 10 . Whilst the use of such a moving filter does impact on the overall spectral resolution on these areas, far more information is retained than using static spectral binning as is enforced by systems with fewer spectral collection channels.”

As mentioned previously it is not the intention of the work to identify the underlying biophysical origins of such small shifts in emission properties, simply to show that they are only observable with our technology. Often shifts are due to phenomena like protein concentration gradients, for example, and so this method would allow quantification of such examples that was not previously possible, however, that is conjecture for future study. It will be clear to anyone who operates in this arena to obtain full understanding of a specific process in a specific sample is highly complex and really only applies to that particular sample. We thus believe that such specialised work is better reported in more specific journals, which is the aim of our current and on-going research as indicated in the text.

It would have been interesting a discussion of the improvement of the full spectral method with 0.5 nm resolution with respect to the hyperspectral imaging in terms of errors in the reconstructed spectra/acquisition time/acquired counts/information extracted from the images.

This is an excellent point and helps to improve our paper. We have increased the discussion around the approach to signal to noise and associated lifetime calculations and the methods used.

Example from text: “With our high number of simultaneous spectral channels, a 256 x 256 pixel image can require over 33 million lifetime fits and so the computational load of the fitting method is an important consideration. Least squares fitting was chosen to process lifetime calculations for the presented study due to speed of computation. The trade-offs between this method and other analysis techniques are well covered in the literature²². After the dark count rate of the sensor has been subtracted a threshold is applied based on the collected intensity in a specific sensor channel. The threshold for lifetime calculations was set at 10 times the background, i.e. fluorescence events in an image pixel are 10 times the background noise resulting from the detector dark count and any scattering within the optical system. For any sensor channel with any image pixel where the signal was below this threshold a lifetime calculation was not performed and the pixel value set to zero.”

Example from text: “The approach of pixel rejection, or application of a moving filter of 8 pixels (4nm), was used to ensure a signal of >10 background for lifetime calculations. This was applied after correction of signals through subtracting sensor dark count and sensor spectral efficiency response. This approach ensured sufficient photon events in the resulting

decays for fitting errors to be dominated by Poisson noise, however, it should be noted that the fitting model used assumes a single exponential decay. Clearly in regions with perturbed emission or multiple emitting species this will introduce deviation from true lifetime values, however this can be overcome by a more complex fitting approach at the sacrifice of computational speed.”

- Does the bleaching affect their images (both intensity and lifetime maps) when acquired with the maximum spatial and temporal resolution?

Bleaching, and the drive to minimise it (except when specifically desired), is a universal effect in all fluorescence imaging techniques. As noted in the introduction, the efficiency with which we collect the data helps to minimise effects such as bleaching. Indeed, the flexibility of our system enables the user to trade off signal to noise against bleaching or other light induced effects through compared to other systems.

Example from text: “Some commercial lifetime imaging systems, such as the Leica Stellaris, can achieve increased spectral resolution through multiple rapid sequential image captures, with incremental shifts in detection wavelength. However, the requirement for multiple image acquisitions slows down the entire acquisition process and contributes greatly to detrimental effects such as photobleaching and loss in image quality due to sample motion.”

We believe that the presented solution provides the optimal approach to collecting maximum time-resolved spectral information whilst minimising effects such as bleaching. It is well known from the literature that excessive bleaching can lead to shortened observed lifetime, this would be the same for any system.

Which is the effect of the z depth into tissue on the spectral and lifetime images? Are their 0.5 nm spectral and 50 ps temporal resolution affected?

Text has been added to the text to address this point,

Example from text: “For fresh tissue samples, there is potentially significant depth penetration of excitation light and therefore collection from multiple species, however, the diffraction limited confocal volume probed is small relative to structures of interest and is not deemed to have a significant perturbation on the signals collected. Similarly, the de-scanned confocal nature of the system causes minimal disruption to the spectral signal as the spectrometer is effectively decoupled from the image scan, and the optical path-length differences due to axial penetration are too small to influence the optical timing response.”

Moreover, the authors reported only static images but it would have been interesting to show also if their system is able to characterize and follow dynamic events. It is important in my opinion at least a discussion about the eventual limitation in the number of spectral channels that can be exploited and the errors in the computed lifetime in dependence on the temporal event under investigation.

The demonstration of increased framerate imaging, through the use of dimensional binning, has been removed for clarity as discussed earlier. It should be noted that the reviewed

version of the paper did indeed show a dynamic system being monitored at 10 fps albeit with reduced resolution. An expanded discussion around image acquisition times has been added. The aim is to report results and the potential of the system. If we explored every avenue that the instrument is capable of the paper would be excessively long, so we have been selective, which I am sure the reviewer appreciates.

Example from text “The goal of this work was to demonstrate the underlying capabilities of the novel system to acquire an unprecedented level of spectral and temporal information. Whilst image capture and histogram formation occur in real-time there is a bottleneck to sequential imaging speed due to data transfer of the spectral data from the FPGA to the PC (currently via USB 3 connectivity). The theoretical maximum size for a 256 x 256 pixel image captured with 512 spectral and 16 (chained) temporal bins is 16Gb. This can lead to a delay of up to 30 seconds in the worst-case scenario in displaying the image. In practice the image size is dependent on the density of sample in the image and the breadth of the emission spectra involved, reducing the image size and associated transfer times. There is potential to reduce this data retrieval time by optimizing of the data link to the PC to enable real time imaging at 0.2 frames per second. As presented in Figures 1 and 2 a high spectral contrast color image can be obtained through three channel (typically RGB) color-mapping requiring only 3 data points per pixel. Since this can be performed on the FPGA before transmission to the PC it represents a possible solution to maintaining spectral contrast whilst requiring low data flow. Clearly if full spectral lifetime information is required the entire, 512 spectral channel, time resolved, dataset must be transmitted to the PC. Here the inherent flexibility in the design of the core architecture comes to the fore allowing spectral and temporal binning to be applied through minor changes in software choices to increase frame rates before transmission to the PC. This enables imaging at up to 10 frames per second (at 128 x 128 pixels). The application of the system to high frame rate imaging utilizing dimensional binning is the scope of future work. The approach of pixel rejection, or application of a moving filter of 8 pixels (4nm), was used to ensure a signal of >10 background for lifetime calculations. This was applied after correction of signals through subtracting sensor dark count and sensor spectral efficiency response. This approach ensured sufficient photon events in the resulting decays for fitting errors to be dominated by Poisson noise, however, it should be noted that the fitting model used assumes a single exponential decay. Clearly in regions with perturbed emission or multiple emitting species this will introduce deviation from true lifetime values, however this can be overcome by a more complex fitting approach at the sacrifice of computational speed. ”

- How many counts they used to compute the lifetime? For example, in Figure 2g (and related video) the authors report a count scale in the range [0-150 counts]. Are these counts exploited to extract the lifetime map in Figure 2h? Which is the error associated with each lifetime due to these few counts?

This particular figure has now been removed for clarity as discussed previously, however, the previous response above covers signal to noise and there is a large discussion in the literature regarding best fitting methods and required photon counts, the choice of method has been added:

Example from text: “With our high number of simultaneous spectral channels, a 256 x 256 pixel image can require over 33 million lifetime fits and so the computational load of the fitting method is an important consideration. Least squares fitting was chosen to process lifetime calculations for the presented study due to speed of computation. The trade-offs between this method and other analysis techniques are well covered in the literature²². After the dark count rate of the sensor has been subtracted a threshold is applied based on the collected intensity in a specific sensor channel. The threshold for lifetime calculations was set at 10 times the background, i.e. fluorescence events in an image pixel are 10 times the background noise resulting from the detector dark count and any scattering within the optical system. For any sensor channel with any image pixel where the signal was below this threshold a lifetime calculation was not performed and the pixel value set to zero.”

Moreover, could the authors explain why there are some black pixels (0 ns lifetime) in the lifetime map associated with non-zero counts pixels in the intensity map and viceversa?

As above this is explained in the text and we are sorry if this was not clear enough previously. The absence of data in lifetime images is from regions where signal to noise was not great enough to insure reliable lifetime calculation.

Example from text: “For any sensor channel with any image pixel where the signal was below this threshold a lifetime calculation was not performed and the pixel value set to zero.”

-Regarding Figure 3: I think the authors show a nice application, especially since they exploit the autofluorescence signal of the sample. However, I think that more information could have been inferred from the data. For example, from the spectral data is it possible to retrieve any differences among pixels by mapping the ratio between the red and green part of the spectrum (above-to-below 600 nm) instead of the mean wavelength?

The figure already shows full spectra for different regions along with the true colour image, both of which show more detail than a simple ratiometric method would exhibit. The splitting of the spectrum into two parts we believe would miss the purpose of a fully spectral system (although the system could perform that task, at significantly increase frame rate, as presented in the original submission, upto 10 frames per second). There is plenty of literature on spectral similarity algorithms that could be employed against the datasets, scope for future adoption and out-with the core message presented. The reviewer has therefore highlighted an area in which the instrument has huge potential and these areas will be explored in future publications.

Are the differences in the lifetime distributions due to different autofluorescent proteins in the samples or to a single protein exposed to different environments?

As previously, the detailed understanding of the origin of the optical signatures is not the core message of the work, a supposition of the most likely origin of the changes across the sample is given backed up by the available literature around both spectral observation and

known effects of mechanisms, such as increased crosslinking on lifetimes, the power of the combined information is in increased contrast across such samples that exhibit subtle emission changes.

Example from text: “The most likely explanation for the observed spectral and lifetime changes across the sample being areas of increased stiffness that have been shown to red-shift the autofluorescence emission wavelength²⁶ due to increased sclerotization (increased cross-linking of proteins²⁷). The lifetime reduction is most likely caused by increasing rates of non-radiative energy loss through self-quenching and local energy transfer to the cross-linked protein matrix. The combination of lifetime and spectral information allows for the subtle changes in the protein matrix to be observed. Whilst a detailed analysis of the underlying biological makeup of the sample is not the purpose of this work it can be clearly seen that full spectral lifetime imaging provides an exceptionally powerful tool for the study of the complex composition of biological systems.”

The authors should at least quantify the differences in the sample characterization that can be obtained by exploiting the spectral method, the lifetime imaging or their combination. Moreover, the authors did not report the errors neither the number of samples analyzed.

We are sorry if this was not present before. We have added text along with clarification that specific sample characterization is not within scope of this work, intended as in introduction to a wide audience of the technology. We do however indicate the type of work that can be undertaken with the system on the specific samples the reader may wish to investigate.

Example from text: “the autofluorescence spectral emission shows subtle variations between pixels, with small changes of only a few nanometres apparent, which would be frequently missed with traditional spectral imaging systems with significantly fewer spectral channels”

Example from text: “whilst a detailed analysis of the underlying biological makeup of the sample is not the purpose of this work it can be clearly seen that full spectral lifetime imaging provides an exceptionally powerful tool for the study of the complex composition of biological systems.”

In the same context, this work is intended as an introduction to the technologies ability to acquire optical signals of unprecedented detail, of interest across a wide range of fields. Comparison of signals to determine, for example, cancerous vs non cancerous tissue signatures which requires large sample sizes and a detailed study of the biological origins of the signal would be targeted at a much more specialist audience.

In my opinion it is not clear what improvements can be gained in the sample

characterization obtained by means of their system with respect to other available spectral/lifetime setup. The authors should have stressed more the novelty of their system and the applications.

In the revised version, stressing the novelty of the system is now paramount and was perhaps lost previously in our enthusiasm for the potential of the system, which both reviewers also clearly share. As mentioned in the previous responses the introduction has been expanded to enhance comparisons with other available systems or those in the literature, notably over and above the increased spectral detail, which we believe the presented data does show, without the need to go into detailed sample characterization.

Along with increased time resolved spectral resolution the presented technology offers significant advancements over previous work in the areas of acquisition speed of this level of detail, through data reduction, and importantly the level of integration allows for easy adoption of the technology without the need for an advanced well equipped optical laboratory.

- In Figures 4 and 5 the authors did not report any error analysis in their graph nor the number of samples.

As above, discussion around errors and the approach to noise has been increased in the main text.

Also in these applications, a more in depth analysis of the acquired data should have been reported. For example, is it possible to extract some interesting information by exploiting only the spectral images or the lifetime maps?

Again the reviewer is clearly excited by the real potential of the system but real detailed analysis, as mentioned before, will be the subject of more specific sample focused publications. However, we fully appreciate the point being made and thus Figure 4 has been updated to have a more refined analysis of the displayed data with close linking to the gold standard histology image shown. We believe this now clearly shows the power of combined lifetime and spectral imagery. Three regions of interest, based on different areas of the H&E have now been selected and both the intensity and lifetime spectra shown instead of just the lifetime spectra for much larger regions that were shown in the previous submission. It is clear that, in some cases spectral information provides the best discriminator and in others lifetime, the combination allows for increasingly robust distinction between cellular regions.

Which is the improvement of using the combined information related to both techniques?

As above the updated figure now clearly shows both spectral intensity and lifetime for different cellular regions clearly showing the power of a combined approach to discriminate between regions. The main text has been updated to increase discussion on this point.

Example from text: "Whilst the intensity spectrum of the emission from each of the regions is of similar shape with only a small shift in the location of the spectral peak (Figure 4 (d),

left) there is far greater contrast in the spectral lifetime response. The shift in Eosin lifetime is likely to be due to a multitude of factors including variation in cellular uptake, with high concentrations leading to common lifetime effects such as self-quenching and inter/intra cellular variations affecting local pH and viscosity. Different pH environments have been shown to affect the emission properties of Eosin such as a red-shift in the spectra, as observed here, and reduction in emission intensity, commonly associated with quenching, resulting in reduced lifetime²⁹. Furthermore the presence of the hematoxylin stain has been shown to increase the observed Eosin lifetime where co-staining of cells occurs²².”

Example from text: “The same three areas of interest as before are shown co-located in Figure 4(f) showing significant spectral intensity differences between the regions, with region three (***) located in the most cancerous area exhibiting a red-shifted emission consistent with previous reports¹². The spectral lifetime also shows distinctly different responses from the three regions with a reduction in lifetime observed as cellular density and increasing levels of amorphous tissue normally associated with cancer manifest¹⁴⁻¹⁷. The collection of both spectral and lifetime information that the presented system enables, with extremely high levels of detail, allows for increasingly robust distinction between cellular regions with some species showing large spectral variation with little lifetime change, and others showing distinct changes in lifetime for a simpler spectral response.”

And again, the improvement of their setup with respect to others should have been stressed more also in these applications. According to which parameter did the authors classify the cancer and healthy ROIs?

As previously the novelty of the application to cancer determination is not the primary message of the work, it is a presented example of an area where high resolution spectral lifetime has the ability to distinguish cellular types with the combination being more powerful than either individually. It has been noted in the text that whilst the gross classification of the sample as cancerous moving through into transitional tissue was made by a pathologist the ROI were chosen for differing cellular type.

Are the differences in the spectral lifetime between cancer and healthy ROIs related to different cellular populations among these areas?

The updated figure and text now show much tighter regions of interest clearly from different cellular populations as shown in the H&E image and also a postulate, based on previous publications with less sophisticated data, on the reasons for such changes.

A more refined comparison should be performed between the spectral/lifetime data and the H&E images. Moreover, is it possible to extract a parameter that allows to identify tumor areas among healthy regions in the acquired spectral/lifetime images and compare the results with those obtained by a pathologist?

As above the figure and text have been updated to link the shown signatures to the H&E image. The determination of cancer vs non-cancerous and potential subtyping is the scope of a large study directed towards that field and was not the goal of this work, this is very much an exemplar, and this is now made clear in the text.

Minor points:

-Figure 3b: the counts scale is missing in the lifetime histograms

The figure has been updated to include this.

-Lines 69-70: "A color image was produced by using an intensity weighted transparency alpha channel to modulate pixel saturation.". It is not clear (at least to me) the procedure followed by the authors to process the images.

Have added detail in the methods section.

Example from text: "A standard intensity weighted transparency was used to modulate pixel saturation (see methods) "

Example from text: "Color RGB values for color images were processed as described in the main text. After RGB values were obtained, the saturation of each pixel was adjusted based on the corresponding overall intensity for that pixel. This was performed by setting the image background to black and adjusting the pixel transparency value, scaled by its normalized intensity."

We hope that the above comments sufficiently address the concerns raised by the reviewers.

REVIEWERS' COMMENTS

Reviewer #2 (Remarks to the Author):

I would like to thank the authors for their answers. I think they properly addressed all the points raised, improving their manuscript.